# ROBOTS ASK THE WAY: COMMUNICATION-ENABLED SOCIAL NAVIGATION

## ABSTRACT

Assistive autonomous robots operating in multi-agent environments require efficient strategies to locate specific individuals among multiple residents. Current social navigation methods focus on reactive collision avoidance and trajectory adaptation, but lack mechanisms to proactively gather information through human-robot communication.

We introduce Communication-enabled Social Navigation (CommNav). In this novel task, robotic agents actively seek assistance from residents to locate target individuals by requesting information about recent sightings, locations, and movement patterns. To evaluate CommNav, we extend Habitat 3.0 to create Habitat 3.0c, a communication-enabled variant supporting multi-human environments with structured information exchange protocols. Adding COMM to a state-of-the-art social navigation model yields an improvement of 7% in finding a specific individual and 8% in overall episode success.

Our experiments reveal that: (i) explicit human-robot communication substantially enhances multi-person navigation performance; (ii) pre-training COMM on a communication pretext task effectively addresses the challenge of occasional interaction signals.

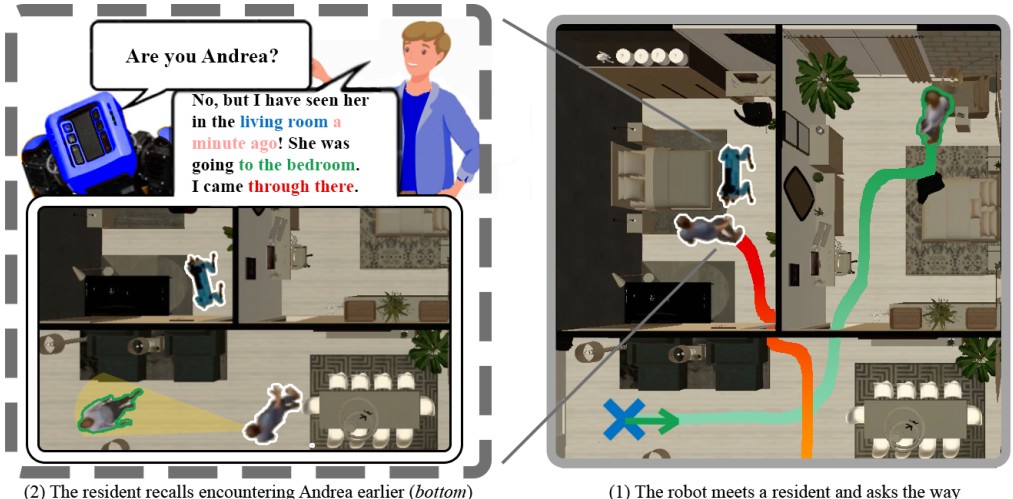

(2) The resident recalls encountering Andrea earlier (*bottom*) and tells the robot (*top*)

(1) The robot meets a resident and asks the way

Figure 1: In Communication-enabled Social Navigation (**CommNav**), the robot actively seeks guidance from residents. If the resident is not Andrea (*left panel*), they may still help by reporting whether they `have seen` ($x_h$) her, `when` ($x_t$) she was last seen, her `location` ($\mathbf{x}_l$), her `direction` ($\mathbf{x}_d$), and their own `path` ($\mathbf{x}_p$). These signals are encoded as structured vectors that condition the navigation policy (see Sec. 3.1).

## 1 INTRODUCTION

Social navigation requires robots to avoid obstacles, interpret human behaviour, and adapt to dynamic, partially unknown environments. Existing approaches have primarily emphasized collision avoidance and motion planning as core solutions Che et al. (2018), yet they lack proactive mechanisms for acquiring task-relevant information from other agents. This limitation becomes particularly critical when robots are required to identify and interact with specific individuals in shared spaces, such as households, offices, or care facilities. For instance, a robotic assistant tasked with delivering medication to a designated person would, under traditional methods, rely on exhaustive room-by-room searches, which is both disruptive to residents and computationally expensive. In contrast, humans naturally employ communication to obtain spatial cues, asking questions such as *"Have you seen Andrea?"* to gather directional guidance or recent location data Pramanick et al. (2024); Daniel & Denis (1998); Sen (2024). Prior work shows that dialogue improves cooperative navigation Amoozandeh et al. (2024); Rehrl et al. (2009), yet this communicative capability remains largely underexplored in the social navigation literature.

Current state-of-the-art social navigation methods Puig et al. (2023); Scofano et al. (2024) focus primarily on trajectory prediction and collision avoidance in multi-agent environments. However, these approaches suffer from several critical limitations when deployed in realistic household settings: they cannot leverage human knowledge of recent spatial events, and they fail to exploit the collaborative potential inherent in human-robot cohabitation scenarios. Incorporating communication into social navigation introduces challenges related to the sparsity and unpredictability of interaction signals, the need to encode heterogeneous information such as temporal, spatial, and directional cues, and the integration of communication that may range from precise coordinates to vague natural language descriptions. Moreover, navigation policies must remain effective and scalable as the number of human agents increases, adding further social complexity.

We introduce Communication-enabled Social Navigation (**CommNav**), a novel task that enables robotic agents to actively seek assistance from residents to locate target individuals through information exchange (See Figure 1). We formalize CommNav as a multi-agent navigation problem where robots can query human agents for spatio-temporal information about target locations. This extends traditional social navigation by incorporating collaborative information-gathering mechanisms. Suppose a robot is tasked with finding a specific person in a home setting to assist them. Instead of searching every room, it could approach nearby residents and ask, "Have you seen them?" The resident might respond with helpful information, such as "I saw them entering the kitchen a few minutes ago," [1] allowing the robot to navigate more efficiently and strategically, reducing the need for exhaustive searching.

To demonstrate the role of communication in social navigation, we introduce a novel component, which we call the **COMM** module, into the leading technique Distributed Proximal Policy Optimization (DDPPO) Wijmans et al. (2020). COMM employs a novel pre-training strategy on a communication pretext task Liang et al. (2019) to address the challenge of learning from infrequent interaction signals, achieving significant performance improvements over non-communicative baselines. A challenge in communication-enabled navigation is bridging the gap between structured mathematical protocols and natural human interaction. While our technical framework supports precise spatio-temporal communication (e.g., coordinates, timestamps, directional vectors), we leverage Large Language Models Yang et al. (2025) to render simulator interactions more human-like, with only minimal degradation in performance. This finding highlights promising directions for future research at the intersection of natural language processing and robotics.

To evaluate this task, we develop **Habitat 3.0c**, a communication-enabled variant of Habitat 3.0 Puig et al. (2023) that supports multi-human environments with realistic collision avoidance (ORCA integration van den Berg et al. (2011)) and structured communication protocols. Through extensive experiments in 12 diverse household environments, we demonstrate that COMM achieves 78% Finding Success compared to 71% for baseline methods, and 24% Episode Success versus 16% for standard approaches. These results establish the first quantitative benchmark for communication-enabled social navigation and validate the effectiveness of collaborative information-gathering approaches in multi-agent environments.

---

[1]This work considers various degrees of human accuracy, e.g. the cue may also read: "I saw them 7 meters North, heading East, 43 seconds ago." See Sec. 3.1 for details.

## 2 RELATED WORK

### 2.1 EMBODIED NAVIGATION

Recent progress in embodied navigation has been fueled by extensive 3D indoor datasets Chang et al. (2017); Shen et al. (2021); Ramakrishnan et al. (2021) and advanced simulators Savva et al. (2019); Shen et al. (2021); Kolve et al. (2017). However, most conventional simulators represent humans as static objects or basic obstacles Xia et al. (2020); Yokoyama et al. (2022), limiting research that requires realistic, dynamic human-robot interactions.

More recent simulators now support more lifelike human-robot interactions. For instance, Habitat 3.0 Puig et al. (2023) and similar systems Li et al. (2024); An et al. (2023) support complex human dynamics, facilitating more nuanced studies of Social Navigation. Habitat 3.0, in particular, allows for advanced Social Navigation and Social Rearrangement scenarios, where agents interact with humans in more realistic settings.

Owing to these advancements, embodied AI (EAI) has expanded to include various tasks, such as PointGoal Navigation Wijmans et al. (2019), ObjectGoal Navigation Batra et al. (2020), and Vision-and-Language Navigation (VLN) Anderson et al. (2018); Krantz et al. (2020). However, most approaches Scofano et al. (2024); Chaplot et al. (2020); Ramakrishnan et al. (2022) focus on static environments based on a single agent, lacking suitability for scenarios where agents must continuously adapt to the presence and movement of humans.

Our proposed task CommNav, goes beyond traditional navigation by incorporating active collaboration between robots and human agents in dynamic and multi-human social settings.

### 2.2 MODELING DYNAMIC SOCIAL ENVIRONMENTS

Simulated environments such as iGibson Li et al. (2021), SEAN Tsoi et al. (2022), VirtualHome Puig et al. (2018), and Habitat 3.0 Puig et al. (2023) provide privileged information about scene and agent elements, which is valuable for training reinforcement learning (RL) policies. In social settings, this privileged information plays a crucial role in predicting human behavior Kivrak et al. (2021) and can assist the robot in more efficiently locating other agents through targeted communication.

Forecasting human trajectories enhances navigation policies Patle et al. (2019). However, in social environments, communication cues are often sparse—such as when a robot must locate one human to ask for help finding another—requiring the system to manage limited social information.

To address the need for efficient predictions in dynamic social environments, we draw inspiration from asynchronous state-action history identification methods Kumar et al. (2021); Loquercio et al. (2023); Zhang et al. (2023); Scofano et al. (2024). Unlike prior work, which seeks to infer state-action mappings in a separate second stage of training, we integrate a pre-trained module directly into an online policy. This approach enables the policy to utilize sparse social cues more effectively, overcoming information gaps in social interactions.

### 2.3 SOCIALLY-AWARE NAVIGATION IN SIMULATED ENVIRONMENTS

Research on socially-aware navigation aims to enable robots to navigate shared spaces while respecting human social norms. Classic methods in multi-agent collision avoidance, such as ORCA and CADRL Berg et al. (2011); Van den Berg et al. (2008); Chen et al. (2017b), have been extended to include social rules Chen et al. (2017a); Ferrer et al. (2013). The socially-aware extension of CADRL (SA-CADRL) Chen et al. (2017a) integrates commonsense social protocols to reduce navigation uncertainty and mitigate the Freezing Robot Problem Trautman & Krause (2010), a common issue where robots halt due to indecision in crowded environments. In addition to collision avoidance, several studies explore more complex human-agent interactions through techniques such as spatiotemporal graph modeling Lu et al. (2022) and social attention mechanisms Chen et al. (2019). These models often rely on fully known environments with predefined human trajectories and simple obstacles Chen et al. (2017b; 2019).

We adopt state-of-the-art techniques in social navigation Wijmans et al. (2020); Scofano et al. (2024), augmenting the reinforcement learning framework with direct communication between the

robot and human agents. We extend the navigation policy with a dedicated communication module, enhancing the robot's Social Awareness when navigating uncharted environments.

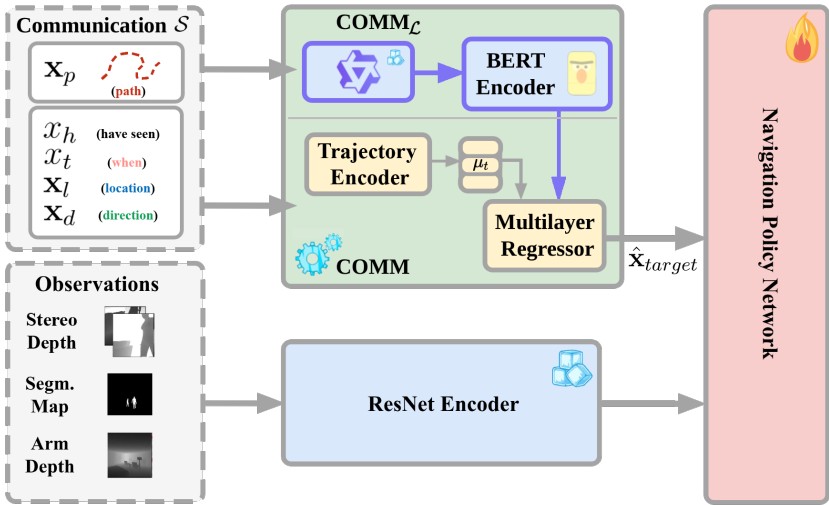

Figure 2: Architecture of COMM model. In the language-enabled COMM$_\mathcal{L}$, $\mathcal{L}$ is obtained by feeding communication data $\mathcal{S}$ to QWEN3-8B. $\mathcal{L}$ is then encoded by a frozen BERT Encoder and used to train a Multilayer Regressor to regress the target position $\hat{\mathbf{x}}_{target}$. The COMM module, on the other hand, directly encodes $\mathcal{S}$. The trajectory $\mathbf{x}_p$ is embedded through a Trajectory Encoder and later combined with $[x_h, x_t, \mathbf{x}_l, \mathbf{x}_d]$ to regress $\hat{\mathbf{x}}_{target}$. This prediction fused with ResNet-encoded observations, guides the Navigation Policy.

## 3 COMMUNICATION-ENABLED SOCIAL NAVIGATION

We introduce Communication-enabled Social Navigation (CommNav), where a robot actively seeks and follows human guidance in real-world environments. Unlike traditional social navigation, which emphasizes collision avoidance and social norms, CommNav leverages communication to improve efficiency and cooperation. By asking for and interpreting directions, robots can reduce exploration time and adapt their behavior in crowded or unfamiliar spaces Burgoon et al. (1995); Joo et al. (2017); Trabelsi et al. (2019); Doherty & Laidre (2022).

In this setting, multiple humans may have observed the individual of interest and are willing to help. The robot does not target a specific person but opportunistically collects cues from anyone it encounters. Once it locates the individual and receives consent, it begins providing assistance. CommNav thus extends social navigation with communication-driven interaction, emphasizing adaptation in dynamic, multi-agent environments Trabelsi et al. (2019); Wu et al. (2021); Šajina & Ivasic-Kos (2023); Liang et al. (2019).

### 3.1 CONTENT OF COMMUNICATION MESSAGES

Formally, when the robot is close to and oriented toward a human (i.e., within its field of view), it receives a message encoded as a state vector $\mathcal{S}$ with the following components:

- $x_h$ (`have seen`), a boolean indicating whether the speaker has observed the target.

- $x_t$ (`when`), a scalar counting steps since the target was last seen;

- $\mathbf{x}_l$ (`location`), the last known 3D location of the target agent $\mathbf{x}_l \in \mathbb{R}^3$;

- $\mathbf{x}_d$ (`direction`), a 3D point $\mathbf{x}_d \in \mathbb{R}^3$ representing the direction of the target human;

- $\mathbf{x}_p$ (`path`), a list of 100 3D coordinates $\mathbf{x}_p \in \mathbb{R}^{100 \times 3}$ representing the trajectory of the speaking agent over the past 100 steps.

All coordinates are expressed relative to the speaking agent's current position, yielding an egocentric description of the encounter.

If the target agent is not in sight, the communication sensor updates only $\mathbf{x}_p$ and $\mathbf{x}_t$, provided the target has been previously encountered. When the target agent is encountered, the human updates $\mathbf{x}_l$ and $\mathbf{x}_d$; at this point, $\mathbf{x}_h$ becomes *True*, and $\mathbf{x}_t$ begins updating. If the agents meet again, all encounter-related variables are overwritten.

Whereas real human communication is approximate and often incomplete, $\mathcal{S}$ provides a precise and exhaustive representation of target encounters. This structured information serves as a foundation for generating natural language instructions. To close the gap with real-world scenarios we use the QWEN3-8B model Yang et al. (2025) to translate each state vector into a short utterance $\mathcal{L}$. To achieve this objective, we use the following system prompt:

```
System Prompt

Inputs:
x_h (seen Andrea?), x_t (time since), x_l (last 3D location),
x_d (Andrea's direction), x_p (last 100 3D steps, current = 0,0,0)

Task:
Return one friendly sentence
  If x_h=True:  report Andrea's distance, direction, heading, and
  your past path.
  If x_h=False:  say you haven't seen Andrea, but report your past
  path.

Constraints:
Use meters, directional terms only ("in front of me," "behind me,"
"to my left/right"), concise, no place names.

Example:
< Example S > "I saw them about 4 meters in front of me and a little
to my right; they were heading to the left.  I came from behind
me."
```

The following example illustrates the outcome of this conversion into a more natural language exchange.

**Ex. 1:** *Input $\mathcal{S}$:* $x_h = 1$, $x_t = 53$, $\mathbf{x}_p \in \mathbb{R}^{100 \times 3}$. *Output $\mathcal{L}$:* "I saw them about 4 m in front of me and slightly to my right; they were heading left. I approached from behind.";

**Ex. 2:** *Input $\mathcal{S}$:* $x_h = 0$, $\mathbf{x}_p \in \mathbb{R}^{100 \times 3}$. *Output $\mathcal{L}$:* "No. I haven't seen them. I came from about 6 m behind and half a metre to my left."

CommNav therefore evaluates not only whether communication improves navigation, but also how effectively structured representations can be grounded into a more natural interaction. Implementation details are provided in the Supplementary Material.

## 3.2 RL WITH HUMAN-ROBOT COMMUNICATION

Our approach, illustrated in Figure 2, extends a standard ResNet-based navigation architecture with a dedicated **Communication Module (COMM)** that provides a parallel stream of guidance. The baseline ResNet encoder processes raw observations (RGB-depth) into a compact representation. In parallel, the COMM module interprets intermittent human-provided inputs ($\mathcal{S}$) and produces an estimated target location, treated as an additional sensor.

The **Navigation Policy Network** then fuses the two streams to generate the actions. When human input is unavailable, the COMM module substitutes a placeholder, and it allows the agent to fall back on its visual policy. This design enables occasional use of human guidance while maintaining complete autonomy.

**Policy Training.** As for state-of-the-art models Puig et al. (2023); Scofano et al. (2024), the navigation policy follows the DDPPO framework and is trained with reinforcement learning objectives

tailored to CommNav, including success-based and efficiency-weighted rewards. Training begins without communication, ensuring the policy learns robust navigation from visual inputs alone.

**Communication Module.** The COMM module is pre-trained to interpret infrequent human-provided information. Rather than concatenating communication data with visual features, it translates cues into an estimated target position $\hat{\mathbf{x}}_{target}$, analogous to a PointGoal sensor Zhao et al. (2021); Partsey et al. (2022).

We pre-train COMM on the proxy task of predicting the target's current position from the interaction message. For this purpose we collect a dataset of $2.4\,\mathrm{M}$ communication instances of $\mathcal{S}$ gathered over 60 million simulation steps of baseline training; each recorded communication instance contains $\mathcal{S} = \{x_h, x_t, \mathbf{x}_l, \mathbf{x}_d, \mathbf{x}_p\}$ together with the ground-truth target location $\mathbf{x}_{target}$, which serves as the regression target. Because the elements of $\mathcal{S}$ are heterogeneous, we process them separately and train COMM in two complementary ways. For the input path, the trajectory $\mathbf{x}_p$ is embedded by the Trajectory Encoder into a hidden vector $\mu_t \in \mathbb{R}^h$. We concatenate $\mu_t$ with the vector $[x_h, x_t, \mathbf{x}_l, \mathbf{x}_d]$ and feed the result to a Multilayer Regressor that outputs the estimated target location $\hat{\mathbf{x}}_{\text{target}}$.

Separately, to train a language-only variant, denoted COMM$_{\mathcal{L}}$, we convert a subset of the dataset into natural-language instructions: 7000 samples from the collected instances are processed with QWEN3-8B Yang et al. (2025) to generate corresponding natural-language reports. COMM$_{\mathcal{L}}$ is then equipped with a BERT encoder Devlin et al. (2019) and trained to regress $\mathbf{x}_{\text{target}}$ using these textual inputs alone. Both training regimes learn the same regression objective (predicting $\mathbf{x}_{\text{target}}$) but from different input modalities (numerical vs. natural language).

This explicit estimate enables the policy to incorporate communicative hints about the target's location and direction without requiring continuous updates, thereby tightly integrating human input with autonomous navigation.

## 3.3 HABITAT 3.0C

We extend Habitat 3.0 Puig et al. (2023), a state-of-the-art Embodied AI simulator supporting tasks such as Social Rearrangement Szot et al. (2023), Social Navigation Scofano et al. (2024); Campari et al. (2022), and PointGoal Navigation Partsey et al. (2022); Wijmans et al. (2020), into a multi-human setting we call Habitat 3.0c. Here, the robot must identify and assist a target among several agents. Identification is triggered when the robot is within communication range and facing a human, defined as $\phi_T = \langle \mathbf{x}_{robot}, \mathbf{x}_{agent} \rangle$. If the individual is not the target, the robot receives an information set $\mathcal{S}$, analogous to a PointGoal sensor Zhao et al. (2021); Partsey et al. (2022), which can help infer the target's location. Since communication occurs only at encounters and is not guaranteed every episode, the robot must handle sparse and incomplete guidance.

To encourage realistic interactions, Habitat 3.0c integrates Optimal Reciprocal Collision Avoidance (ORCA) van den Berg et al. (2011) and introduces a probability $p = 0.25$ that humans ignore the robot for an entire episode. This modification allows humanoid-initiated collisions, teaching the robot to yield dynamically.

## 4 EXPERIMENTS

Section 4.1 presents results on the newly defined CommNav task, comparing it with the classical single-human SocialNav Puig et al. (2023) and highlighting key differences. Section 4.1 also explores our qualitative results. This includes an analysis of the state-of-the-art solution relative to the proposed Communication-Aware architecture. Section 4.2 presents an ablation study on the number of humans present in an environment.

**Baselines.** We evaluate two top-performing social navigation methods: Distributed Proximal Policy Optimization (DDPPO) Puig et al. (2023), Social Dynamics Adaptation (SDA) Scofano et al. (2024). **DDPPO** is a state-of-the-art reinforcement learning approach that uses a recurrent neural network to generate navigation actions based on egocentric sensory inputs, including stereo depth, a human detector, and GPS coordinates of nearby humans. **SDA** enhances real-time adaptation to human movement. SDA follows a two-stage learning framework: the first stage encodes fully visible human trajectories as social dynamics cues, conditioning the robot's motion policy on this encoded information. The policy operates without direct trajectory access in the second stage, infer-

ring dynamics from the robot's state-action history. In our experiments for Habitat 3.0c, we only use the first stage, which provides an upper bound on performance by leveraging complete trajectory information.

**Metrics.** We use the metrics defined for the Social Navigation task in Puig et al. (2023); Scofano et al. (2024) and introduce one new metric specifically designed for this task. The metrics are as follows: *Finding Success (S)*, the ratio of episodes where the agent successfully located and reached the human; *Finding Success Steps ($S_{steps}$)*, the average steps taken to initially find the human; *Finding Success Weighted by Path Steps (SPS)*, assessing path efficiency relative to the optimal steps to reach the human; *Following Rate (F)*, the ratio of steps where the robot maintains a 1-2 meter distance from the human while facing it; *Collision Rate (CR)*, the ratio of episodes ending with any collision; *Collision Rate with Target ($CR_T$)*, the ratio of episodes ending in a collision with the target human; *Episode Success (ES)*, the ratio of episodes where the agent successfully located and followed the human for the required steps while maintaining a safe 1-2 meter distance. This newly added metric— *Collision Rate with Target ($CR_T$)*—helps evaluate the robot's ability to selectively navigate around its target while avoiding potential collisions with non-target agents, which is crucial for social navigation in multi-agent settings.

**Experiments Setup.** For this study, experiments are conducted following the same training settings discussed in the SoTA social navigation works Puig et al. (2023); Scofano et al. (2024): training is done on the newly proposed Habitat 3.0c simulator on 24 parallel environments, and evaluation on 12 different environments. The experiments used 4xA100 GPUs and a training time of six days for 200M steps. The evaluation is done on a single A100 GPU for four hours.

## 4.1 COMMNAV RESULTS

Table 1: Dialogue-Driven Social Navigation Results. "Comm." specifies whether communication between human and robot is enabled, while "MH" indicates the presence of multiple humans in the environment.

| Models | Comm. | MH | Habitat | S (↑) | $S_{steps}$ (↓) | SPS (↑) | F (↑) | CR (↓) | $CR_T$ (↓) | ES (↑) |
|---|---|---|---|---|---|---|---|---|---|---|
| DDPPO Puig et al. (2023) | - | - | 3.0 | $0.76 \pm 0.02$ | $483 \pm 6.50$ | $0.34 \pm 0.01$ | $0.29 \pm 0.01$ | $0.48 \pm 0.03$ | - | $0.40 \pm 0.02$ |
| SDA Scofano et al. (2024) | - | - | 3.0 | $0.91 \pm 0.01$ | $415 \pm 8.00$ | $0.45 \pm 0.01$ | $0.39 \pm 0.01$ | $0.57 \pm 0.02$ | - | $0.43 \pm 0.02$ |
| DDPPO Puig et al. (2023) | - | ✓ | 3.0c | $0.71 \pm 0.02$ | $618 \pm 1.00$ | $\mathbf{0.40 \pm 0.02}$ | $0.10 \pm 0.01$ | $0.64 \pm 0.02$ | $0.30 \pm 0.01$ | $0.14 \pm 0.02$ |
| SDA Scofano et al. (2024) | - | ✓ | 3.0c | $0.70 \pm 0.01$ | $613 \pm 3.50$ | $0.40 \pm 0.01$ | $0.10 \pm 0.01$ | $0.76 \pm 0.01$ | $0.27 \pm 0.02$ | $0.16 \pm 0.01$ |
| DDPPO Puig et al. (2023) | ✓ | ✓ | 3.0c | $0.70 \pm 0.02$ | $608 \pm 4.00$ | $0.40 \pm 0.01$ | $0.10 \pm 0.01$ | $0.68 \pm 0.03$ | $0.30 \pm 0.02$ | $0.14 \pm 0.01$ |
| COMM | ✓ | ✓ | 3.0c | $\mathbf{0.78 \pm 0.01}$ | $572 \pm 2.00$ | $0.38 \pm 0.01$ | $\mathbf{0.13 \pm 0.01}$ | $\mathbf{0.51 \pm 0.02}$ | $\mathbf{0.23 \pm 0.02}$ | $\mathbf{0.24 \pm 0.02}$ |
| COMM$_{\mathcal{L}}$ | ✓ | ✓ | 3.0c | $\mathbf{0.78 \pm 0.01}$ | $\mathbf{546 \pm 5.00}$ | $0.39 \pm 0.01$ | $0.12 \pm 0.02$ | $0.59 \pm 0.01$ | $0.28 \pm 0.02$ | $0.20 \pm 0.01$ |

Table 1 compares performance in Social Navigation tasks on Habitat 3.0 (top) and Communication-enabled Social Navigation on the extended Habitat 3.0c environment (bottom). To adapt SDA Scofano et al. (2024) for multi-human CommNav task, we enhanced the model to capture the inherent social dynamics and also used DDPPO Wijmans et al. (2020) as a baseline.

Transitioning from a single-human to a multi-human environment causes a significant drop in *Episode Success (ES)*: for DDPPO and SDA, ES falls from 0.40 and 0.43 in the single-human setup to 0.14 and 0.16, respectively, in the multi-human scenario. It highlights the increased complexity in navigating multi-human settings, which require the robot to distinguish among different agents and seek assistance as needed.

The second section of Table 1 shows that incorporating communication into CommNav is non-trivial. Training DDPPO with integrated communication produces no gains (ES remains at 0.14), whereas our proposed COMM achieves an ES of 0.24. This improvement indicates COMM's more consistent ability to successfully complete episodes and handle sparse, complex communicative information. For *Finding Success (S)*, COMM scores 0.78—an 8% improvement over DDPPO with communication enabled (0.70) and respectively 7%, 8% over the re-trained non-communication DDPPO and SDA models. In terms of efficiency, COMM executes the task with an average of 572 *Finding Success Steps* ($S_{steps}$), compared to 608 for Comm.-DDPPO and 618 and 614 for the non-communication baselines DDPPO and SDA, respectively. This improvement indicates a better ability to identify the correct human with fewer steps.

Although the *Finding Success Weighted by Path Steps (SPS)* metric shows a slight decrease (from 0.40 to 0.38) due to additional steps taken when approaching humanoids for guidance, this trade-off contributes to an overall improvement in *Finding Success* (from 0.70 to 0.78) in challenging scenarios. In terms of collisions, COMM achieves a *Collision Rate (CR)* of 0.51, lower than 0.68 for Comm.-DDPPO and 0.64/0.76 for non-communication DDPPO and SDA, respectively. This lower CR indicates that COMM experiences fewer collisions with other agents, suggesting enhanced situational awareness. Finally, COMM's *Collision Rate with Target ($CR_T$)* is minimized to 0.23, underscoring the model's ability to maintain a safe distance from the designated target while closely following the target.

Transitioning from structured communication to generated language instructions, $COMM_{\mathcal{L}}$ achieves a Finding Success of $0.78$ with Ssteps $= 546$, closely matching the performance of COMM. However, episode-level metrics show modest degradation: Episode Success drops from 0.24 to 0.20, collision rate increases (CR: $0.51 \rightarrow 0.59$; $CR_T$: $0.23 \rightarrow 0.28$), Following decreases slightly ($0.13 \rightarrow 0.12$), and SPS changes marginally ($0.39 \rightarrow 0.38$). These results suggest that while the model is largely robust across modalities, the added noise and underspecification of generated language instructions make navigation more challenging and reduce episode-level reliability.

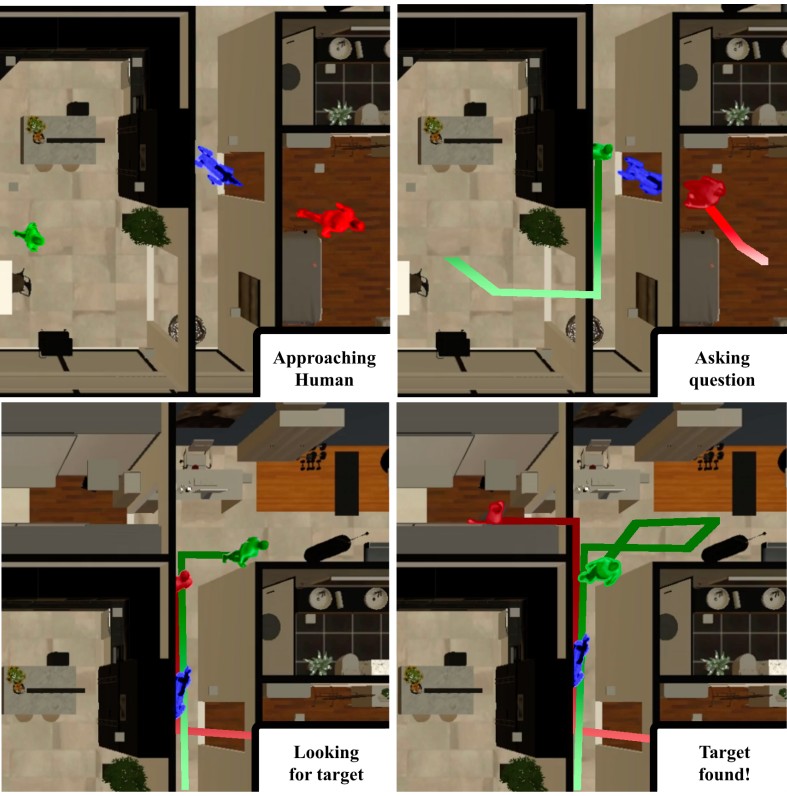

Figure 3: Sequence of human-robot communication for target localization. In frame 1, the robot navigates without input and detects a non-target human. In frame 2, the non-target human signals that it has seen the target human and provides directional cues: when, where, the direction, and the path he took. In frame 3, the robot adjusts its course in response to these cues, aligning its trajectory toward the target. In frame 4, the robot successfully locates and approaches the target human, demonstrating the effectiveness of real-time human guidance in robotic navigation.

Overall, these metric improvements highlight COMM's effectiveness in utilizing its communication module. With support from a simple proxy task, COMM establishes a novel state-of-the-art in communication-enabled social navigation within complex, multi-human environments. Qualitatively, Figure 3 shows how real-time human guidance helps the robot locate the target. After receiving directional cues in frame 2, the robot changes course in frame 3 and successfully approaches the target in frame 4.

## 4.2 MULTIPLE HUMAN AGENTS

Table 2: Ablation: three-human evaluation.

| Models | Comm. | #Humans | $S \uparrow$ | $S_{\text{steps}} \downarrow$ | SPS↑ | F↑ | CR↓ | $\text{CR}_T \downarrow$ | ES↑ |
|---|---|---|---|---|---|---|---|---|---|
| DDPPO Puig et al. (2023) | - | 3 | 0.61 | 730 | 0.34 | 0.07 | 0.80 | 0.27 | 0.07 |
| SDA Scofano et al. (2024) | - | 3 | 0.62 | 708 | 0.35 | 0.07 | 0.81 | **0.24** | 0.08 |
| COMM | ✓ | 3 | **0.67** | 649 | 0.35 | **0.09** | **0.73** | 0.25 | **0.12** |
| COMM$_\mathcal{L}$ | ✓ | 3 | **0.67** | **646** | **0.36** | 0.08 | 0.77 | 0.26 | 0.11 |

Table 2 presents the evaluation of the non-communicative baselines DDPPO Puig et al. (2023) and SDA Scofano et al. (2024), alongside our proposed models, COMM and COMM$_\mathcal{L}$. The evaluation uses three humanoid agents in the same test environments.

COMM and COMM$_\mathcal{L}$ outperforms the non-communicative baselines. Both COMM variants raise *Finding Success* $S$ to 0.67 (vs. 0.62 for SDA and 0.61 for DDPPO) and reduce search effort: $S_{\text{steps}}$ falls to 649 (COMM) and 646 (COMM$_\mathcal{L}$) from 708/730 for SDA/DDPPO. COMM attains the highest *Episode Success* (ES = 0.12) and *Following Rate* (F = 0.09); the language variant nearly matches efficiency (ES = 0.11, F = 0.08) but shows a small drop in SPS (0.33 vs. 0.35 for COMM). Collision rates remain high across models (CR $\approx 0.73 - 0.81$); SDA is marginally better on $\text{CR}_T$ (0.24). In sum, adding communication yields clear gains in success and efficiency, while replacing structured signals with language is viable with a modest trade-off in episode success and path-quality (see Table 2). Overall, while the enhanced performance of COMM can be attributed to the effective incorporation of communication, these results also highlight the inherent challenges of navigating densely populated household environments, where improved safety measures are essential.

**Limitations.** We adopt a structured communication protocol to keep the problem tractable. Real-world conversation, however, is far less formal, with greater variability and ambiguity. Bridging this gap will require training LLMs to express themselves more humanly. Our evaluation is also conducted in simulation with noiseless sensing and perfect egocentric grounding, thus sidestepping the perceptual noise and uncertainty of real environments. Finally, while socially aware navigation holds promise for supporting vulnerable users, it raises ethical risks, particularly around surveillance. In our experiments, interactions are limited to consenting agents, but real-world deployment will demand stronger safeguards, including explicit consent, privacy-preserving data handling, and clear policy constraints on communication.

## 5 CONCLUSIONS

In this work, we introduced Communication-enabled Social Navigation (CommNav), the first framework to explicitly integrate human–robot communication into embodied navigation. By extending Habitat 3.0 into the communication-aware Habitat 3.0c simulator, we enabled more realistic multi-human interactions and richer social dynamics. At the core of our approach is the COMM module, which exploits sparse but informative communicative cues to guide navigation policies. Through both structured simulator-style exchanges and a humanized variant generated by an LLM, our method achieves consistent improvements in success, efficiency, and safety. Our results establish communication as a key mechanism for scaling navigation to complex, crowded, and dynamic social settings.

**Ethical Statement** This work investigates communication-enabled social navigation in simulated multi-human environments. All experiments were conducted entirely within the Habitat 3.0c simulation platform and did not involve real human participants, thereby avoiding risks related to privacy, safety, or consent. The proposed methods are intended to support the development of assistive robotic systems that interact transparently and respectfully with humans. While improved communication capabilities can enhance efficiency in locating individuals, we recognize the importance of ensuring that such technologies are deployed responsibly, with appropriate safeguards to prevent misuse for surveillance or other harmful purposes.

**LLM Usage Statement** To ensure transparency, we note that Large Language Models (LLMs) were employed to revise and refine the writing of the paper, improving clarity and readability.

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

## ADDITIONAL MATERIALS

This supplementary material provides additional implementation details, precise shapes and units for the structured communication vectors used by our task, further architecture details for the COMM module, and extended ablations on (i) training-duration vs. architecture, (ii) robustness to partial communication, and (iii) the role of collision-avoidance (ORCA) and target identification (ReID) modules.

## A  IMPLEMENTATION DETAILS

### A.1  COMMNAV

At every simulation step, human agents register the information they need to communicate to the robotic agent in (1), and (2) :

$$S_{target} = (x_h, x_t, \mathbf{x}_l, \mathbf{x}_d) \in \mathbb{R}^8,$$

$$S_{agent} = (\mathbf{x}_p) \in \mathbb{R}^{(100,3)}.$$

If the agents meet again, all encounter-related variables are overwritten. However, the non-goal agent may inform the robot that they encountered the target agent over $n = 100$ steps earlier; in this case, $\mathbf{x}_p$ would no longer contain a valid path to the encounter location. During simulation, the distance $d_p$ a humanoid agent with linear speed $v = 10.0$ - inherited from Habitat 3.0 Puig et al. (2023) - covers between two steps, averages around 30 centimeters, making the communicated trajectory of distance $|\mathbf{x}_p| = n \times d_p$ centimeters long.

### A.2  COMM MODULE

The TCN Encoder consists of 1D-Convolutional layers with a hidden dimension of $H = 64$, dropout rate $p_d = 0.2$, and ReLU activation. The Transformer Encoder uses an input dimension of $H$, with $h = 3$ attention heads and $l = 2$ layers, while the ST-MLP features a hidden and output dimension of $H$.

## B  ABLATION ON COMM

Table 3 investigates whether the performance improvements of the proposed COMM model result solely from extended training or from its specialized design. To clarify, our training configurations are as follows:

Table 3: Ablation on longer training.

| Models | Comm. | Train Steps (M) | S ↑ | $S_{steps}$ ↓ | SPS ↑ | F ↑ | CR ↓ | $CR_T$ ↓ | ES ↑ |
|---|---|---|---|---|---|---|---|---|---|
| DDPPO Puig et al. (2023) | - | 200 | 0.71 | 618 | 0.40 | 0.10 | 0.64 | 0.30 | 0.14 |
| DDPPO Puig et al. (2023) | - | 270 | 0.72 | 573 | **0.42** | 0.10 | 0.72 | 0.24 | 0.16 |
| DDPPO Puig et al. (2023) | ✓ | 200 | 0.70 | 608 | 0.40 | 0.10 | 0.68 | 0.30 | 0.14 |
| DDPPO Puig et al. (2023) | ✓ | 270 | 0.71 | 618 | 0.38 | 0.11 | 0.64 | 0.28 | 0.16 |
| COMM *(Full Train)* | ✓ | 200 | 0.67 | 652 | 0.38 | 0.10 | 0.71 | 0.254 | 0.18 |
| COMM *(Full Train)* | ✓ | 270 | 0.64 | 659 | 0.39 | 0.10 | 0.65 | 0.18 | 0.18 |
| COMM *(Proposed)* | ✓ | 200+70 | **0.78** | **572** | 0.38 | **0.13** | **0.51** | **0.23** | **0.24** |

In our experiments, we compare several variants, namely DDPPO (the standard baseline trained on the original Habitat 3.0 without any human or communication enhancements), DDPPO with Communication (the baseline trained on the extended Habitat 3.0c environment incorporating multiple humans without communication cues), COMM *(Full Train)* (where the navigation policy is trained from scratch for 200M or 270M steps with the COMM module frozen from the start), and COMM *(Proposed)* (which fine-tunes DDPPO with Communication enabled for an additional 70M steps using the frozen COMM module, demonstrating that integrating communication into an already trained policy can help overcome performance plateaus without the need to train the entire policy from scratch).

For the DDPPO baseline without communication (first and second rows), extending training from 200M to 270M steps yields only marginal improvements—*Finding Success (S)* increases from 0.71 to 0.72, *Collision Rate (CR)* from 0.64 to 0.72, and *Episode Success (ES)* from 0.14 to 0.16. A similar plateau is observed when communication is added in DD-PPO (third and fourth rows). In contrast, the COMM approach shows a clear benefit: while the COMM + Full Train models (fifth and sixth rows) achieve moderate scores (*S* of 0.67/0.64, *CR* of 0.71/0.65, and *ES* of 0.18), the COMM (*Proposed*) variant attains substantially higher performance (*S* of 0.78, *CR* of 0.51, and *ES* of 0.24).

These results indicate that the performance improvements of COMM are not simply due to prolonged training. Instead, they arise from its ability to leverage sparse communication cues. Moreover, by fine-tuning an existing policy rather than training from scratch, the proposed approach reduces training time by approximately six days on a 4xA100 GPU setup, demonstrating the practical benefits of integrating communication into pre-trained policies.

## B.1 SCALABILITY TO MULTIPLE HUMANS

Table 4: Three-humanoid experiments (agents at half speed).

| Models | Comm. | #Hum | Spe. | S↑ | $S_{steps}$↓ | SPS↑ | F↑ | CR↓ | $CR_T$↓ | ES↑ |
|--------|-------|------|------|------|------|------|------|------|------|------|
| COMM | ✓ | 3 | – | **0.67** | **649** | 0.35 | 0.09 | 0.73 | 0.25 | 0.12 |
| DDPPO | – | 3 | ½ | 0.58 | 799 | 0.37 | 0.10 | 0.49 | 0.15 | 0.21 |
| SDA | – | 3 | ½ | 0.61 | 760 | 0.42 | 0.11 | 0.53 | 0.14 | 0.24 |
| COMM | ✓ | 3 | ½ | **0.67** | 711 | **0.45** | **0.13** | **0.42** | **0.11** | **0.29** |

One limitation of Habitat 3.0 is that environments rarely support more than three humanoid agents without severe crowding effects. To simulate denser multi-human settings, we adopt a speed-halving strategy, reducing each humanoid's velocity by 50%. This modification allows more agents to co-exist in the same environment while mitigating blind-spot collisions that would otherwise dominate robot failure cases.

Table 4 reports results in the three-human, half-speed setting. As expected, slowing human motion increases the number of steps required to complete a task, reflected in higher $S_{steps}$ values. However, this trade-off provides more consistent communication opportunities and fewer occlusion-induced errors, ultimately improving overall Episode Success. Notably, our COMM model outperforms baselines even under this more challenging multi-agent configuration, achieving an ES of 0.29 compared to 0.21 for DDPPO and 0.24 for SDA.

These results confirm that communication remains an effective mechanism for scaling navigation performance in crowded environments. Despite increased search times, the COMM module enables the robot to leverage redundant cues across multiple humans, maintaining robust orientation toward the target and improving success rates relative to non-communicative baselines.

## B.2 IMPACT COMMUNICATION COMPONENTS IN COMMNAV

To evaluate the robustness of CommNav to noisy or incomplete communication, we perform ablations over the components of the state vector $\mathcal{S}$: $x_h$ (seen flag), $x_t$ (time since last seen), $\mathbf{x}_l$ (last known location), $\mathbf{x}_d$ (facing direction), and $\mathbf{x}_p$ (trajectory). In each ablation, one component (or a truncated version of $\mathbf{x}_p$) is masked or removed from the input, simulating the effect of partial or noisy communication.

Table 5 reports navigation performance across these ablations. Results show that the COMM module maintains strong robustness: Episode Success (ES) only degrades moderately when individual components are removed, and the model continues to outperform naive or language-only variants. For instance, removing $x_h$ reduces ES from 0.24 to 0.19, while masking $\mathbf{x}_l$ or $\mathbf{x}_d$ yields similar but not catastrophic performance drops. Interestingly, shortening the trajectory history to 20 or 50 steps has minimal impact, suggesting that COMM primarily relies on recent trajectory information rather than long-range history.

These findings confirm that the navigation policy is not brittle to missing or noisy communicative cues. Instead, it leverages redundancy in $\mathcal{S}$ to remain oriented towards the goal. Furthermore,

Table 5: Ablation of Communication Components in CommNav.

| Ablation | S↑ | $S_{steps}$↓ | SPS↑ | F↑ | CR↓ | $CR_T$↓ | ES↑ |
|---|---|---|---|---|---|---|---|
| COMM | 0.78 | 572 | 0.38 | 0.13 | 0.51 | 0.23 | 0.24 |
| $-x_h$ | 0.73 | 595 | 0.37 | 0.11 | 0.58 | 0.25 | 0.19 |
| $-x_t$ | 0.77 | 528 | 0.40 | 0.12 | 0.61 | 0.27 | 0.19 |
| $-\mathbf{x}_l$ | 0.77 | 555 | 0.38 | 0.12 | 0.59 | 0.23 | 0.20 |
| $-\mathbf{x}_d$ | 0.76 | 556 | 0.37 | 0.12 | 0.55 | 0.24 | 0.22 |
| $-\mathbf{x}_p$ | 0.78 | 527 | 0.41 | 0.12 | 0.59 | 0.25 | 0.18 |
| $-\mathbf{x}_p^{20}$ | 0.78 | 517 | 0.41 | 0.13 | 0.60 | 0.27 | 0.21 |
| $-\mathbf{x}_p^{50}$ | 0.76 | 535 | 0.40 | 0.13 | 0.58 | 0.25 | 0.22 |

COMM $\mathcal{S}$, which replaces structured communication with generated utterances, achieves comparable Finding Success (S = 0.78) but slightly lower Episode Success (0.20 vs. 0.24), reflecting the additional noise and underspecification introduced by natural language. Overall, the ablation study demonstrates that CommNav is robust to noise in communication signals, with the COMM module ensuring consistent performance even under degraded inputs.

## C  ABLATION ON OBSTACLE AVOIDANCE AND IDENTIFICATION

In this section, we demonstrate the importance of obstacle avoidance for humanoid agents and target identification for the robotic agent.

**Quantitative Results**

Table 6: Ablation Results on the components of Habitat 3.0c.

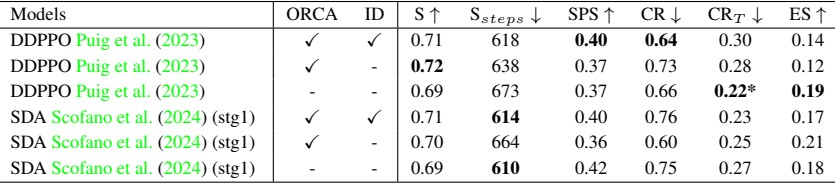

| Models | ORCA | ID | S ↑ | $S_{steps}$ ↓ | SPS ↑ | CR ↓ | $CR_T$ ↓ | ES ↑ |
|---|---|---|---|---|---|---|---|---|
| DDPPO Puig et al. (2023) | ✓ | ✓ | 0.71 | 618 | **0.40** | **0.64** | 0.30 | 0.14 |
| DDPPO Puig et al. (2023) | ✓ | - | **0.72** | 638 | 0.37 | 0.73 | 0.28 | 0.12 |
| DDPPO Puig et al. (2023) | - | - | 0.69 | 673 | 0.37 | 0.66 | **0.22*** | **0.19** |
| SDA Scofano et al. (2024) (stg1) | ✓ | ✓ | 0.71 | **614** | 0.40 | 0.76 | 0.23 | 0.17 |
| SDA Scofano et al. (2024) (stg1) | ✓ | - | 0.70 | 664 | 0.36 | 0.60 | 0.25 | 0.21 |
| SDA Scofano et al. (2024) (stg1) | - | - | 0.69 | **610** | 0.42 | 0.75 | 0.27 | 0.18 |

From a quantitative perspective, removing the identification and collision avoidance components reveals valuable insights into the complexity of the trained navigation policy. Table 6 compares the baseline non-communicative model DDPPO Puig et al. (2023) trained across three different configurations, with and without the identification and collision avoidance components in Habitat 3.0c.

The first model in Table 6 is the baseline non-communicative DDPPO Puig et al. (2023) in Habitat 3.0c. Removing the target identification module (second row), the *Collision Rate* worsens, increasing from 0.64 to 0.73.

We observe a performance shift when the collision avoidance module is ablated (last row of Table 6). Although *Episode Success* (ES) increases slightly from 0.14 to 0.19, *Shortest Path Similarity* (SPS) drops from 0.40 to 0.37, and the *Total Steps Taken* ($S_{steps}$) increases from 618 to 673. These results reflect a more static and inefficient behavior overall.

**Qualitative Results** In a multi-person environment, agents should not physically overlap, and social agents are generally assumed to have recognition capabilities to identify individuals of interest. In Habitat 3.0c, we adhere to these assumptions by integrating collision avoidance logic for humanoids and an oracular identification module for the robotic agent.

Beyond being necessary from a logical standpoint, these modules significantly influence task performance. Figure 4 illustrates the confusion caused by the absence of a ReID module.

Figures 4a and 4b depict the same evaluation episode in environments where the robot either lacks or incorporates a reidentification module.

Figure 4 shows the same evaluation episode in two different environments. Figure 4a depicts the robot trained without the identification module, while Figure 4b shows the robot with the module enabled. At step 150, both robots face an identical scenario: two humanoid agents cross paths, with the non-target agent (wearing a blue shirt) obstructing the view of the target agent (wearing a white shirt). By step 210, the behavior of the two robots diverges. In Figure 4a, the robot incorrectly follows the blue-shirted agent, abandoning the white-shirted target. Conversely, in Figure 4b, the robot correctly tracks the white-shirted target, thanks to the identification module. These qualitative results highlight the critical role of target identification for completing the CommNavtask and in Habitat 3.0c.

Figure 5 demonstrates the importance of collision avoidance logic in humanoid agents. The robot is visible on the left, while the non-target agent is on the right. In this episode, the humanoid agents cross paths again. At step 360, the robot correctly identifies the target agent, aided by the identification module. However, the humanoid agents lack collision avoidance logic, resulting in overlap at step 410. This unnatural interaction causes the robot to mistakenly follow the non-target agent for 420 steps despite initially identifying the correct target.

To further analyze the confusion caused by the absence of these modules, Figures 6 and 7 show the robot's perspective when these modules are removed. Figure 6 presents two views of the same scene: in Figure 6a, the absence of the identification module results in all agents being assigned the same semantic ID, whereas Figure 6b shows the target agent being uniquely identified. Figure 7 shows the robot's view when the humanoid agents overlap due to the lack of collision avoidance logic. In Figure 7a, where the agents overlap, it is difficult to distinguish the foreground agent from the background agent. In contrast, Figure 7b demonstrates that collision avoidance logic resolves this ambiguity, making the scene more natural and agents easier to distinguish.

These results highlight how integrating effective collision avoidance logic (e.g., ORCA Snape et al. (2010)) and a basic identification module reduces confusion and unnatural behaviors in the simulator. These modules are essential for exposing the robot to scenarios that closely mimic real-world conditions, thereby improving the robustness of its navigation policy.

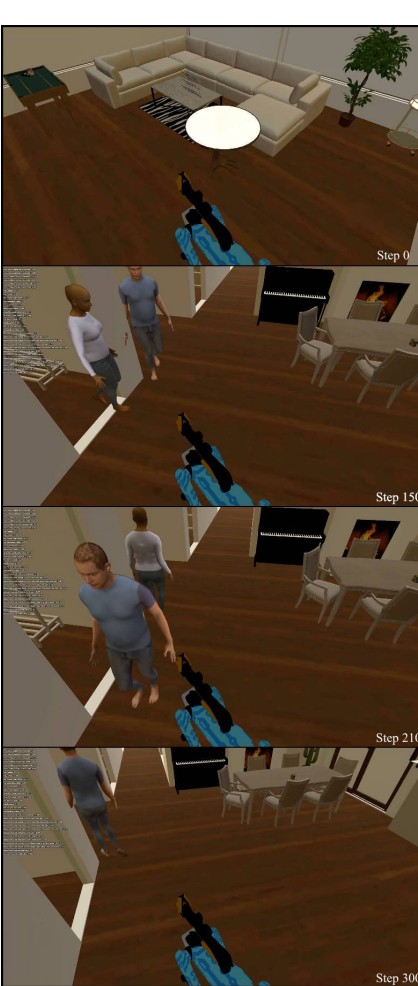
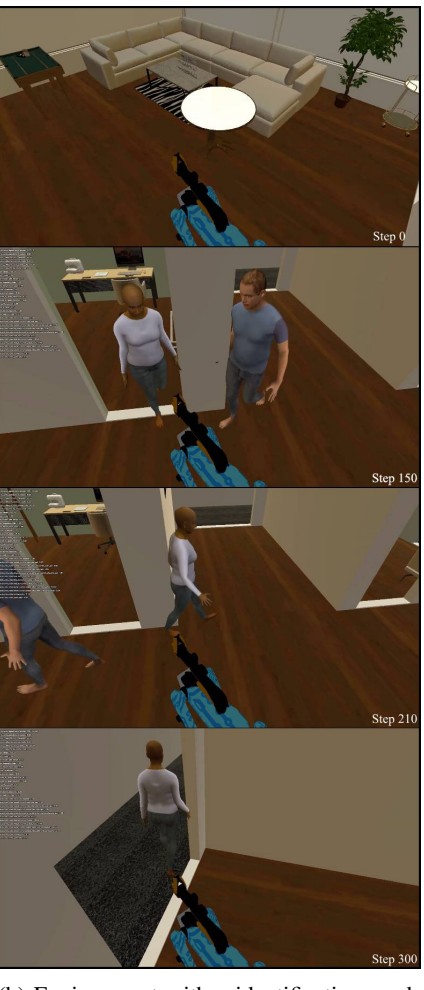

(a) Environment without reidentification module

(b) Environment with reidentification module

Figure 4: Same evaluation episode under two different environments: in the first environment, there is no ReID module in the robotic agent; in the second environment, the robot is equipped with a ReID module.

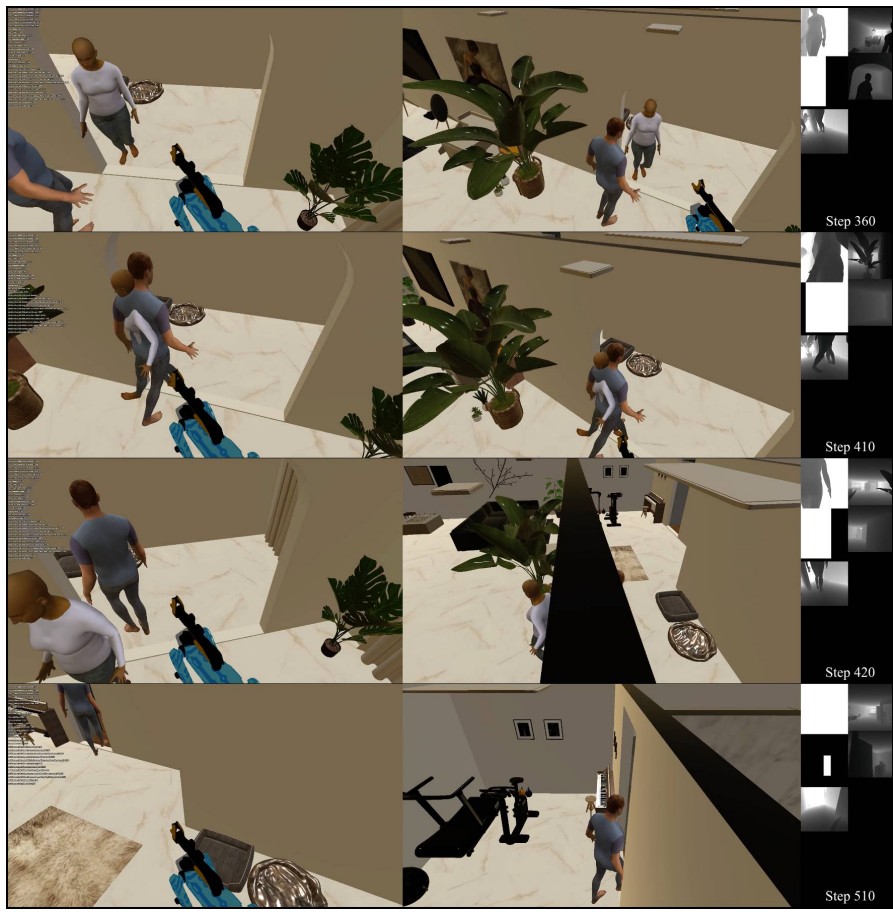

Figure 5: Evaluation episode of the ddppo baseline model without additional modules. Compenetration confuses the robot as it changes the target of its following.

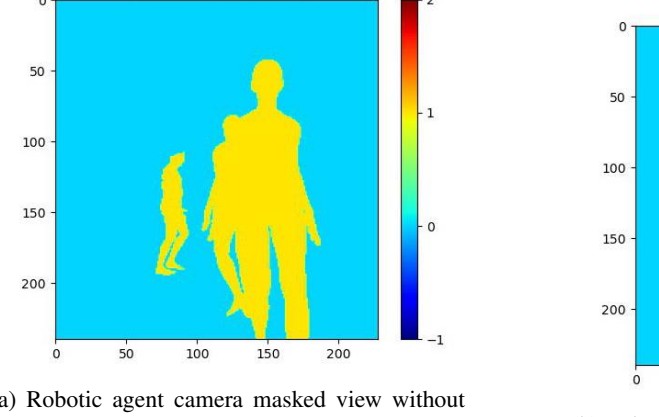

(a) Robotic agent camera masked view without ReID

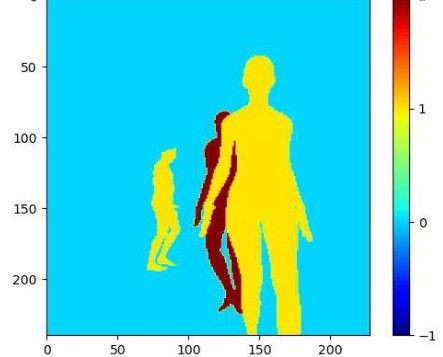

(b) Robotic agent camera masked view with ReID

Figure 6: Segmentation masks with or without target ReIdentification

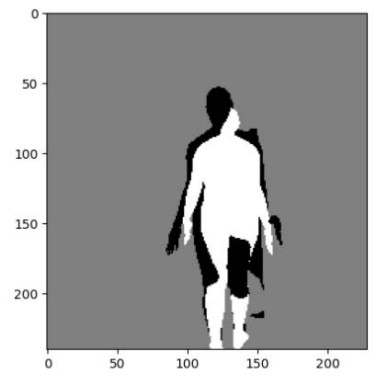

(a) Two agents robotic view with id and no orca

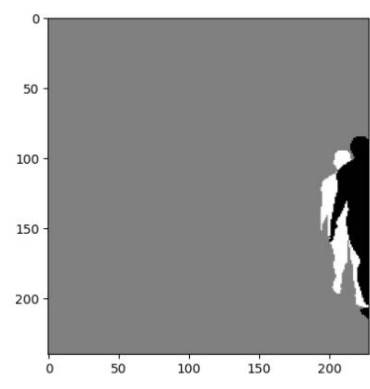

(b) Two agents robotic view with id and orca

Figure 7: Humanoid agents before and after the implementation of ORCA in Habitat 3.0c. (a) shows agents compenetrating. After the introduction of ORCA, agents avoid each other (b).

