# OpenReview forum: "Robots Ask the Way: Communication-Enabled Social Navigation"
_ICLR.cc/2026/Conference — ICLR 2026 Conference Withdrawn Submission_

### Official Review · Reviewer_Gx3c · 2025-10-30

**Soundness:** 3
**Presentation:** 3
**Contribution:** 2
**Rating:** 6
**Confidence:** 3

**Summary:**

Building upon existing LLM-based semantic navigation frameworks, this work introduces a
proactive communication paradigm. They propose and instantiate the CommNav module together
with a continuous-control reinforcement-learning policy in the DDPPO framework. Exploiting
structured messages to perform multi-agent trajectory forecasting and collision avoidance. The
state representation is systematically augmented with communication-aware features, enabling the
agent to reason about both perceptual and linguistic cues.
They extend the Habitat 3.0 simulator into Habitat 3.0c, a multi-human environment that
integrates reciprocal collision avoidance (ORCA) and structured information-exchange protocols.
Experiments quantifying navigation performance, communication efficiency, and collision statistics
demonstrate that the incremental incorporation of the communication module yields consistent
improvements in semantic navigation accuracy and overall task success rate.

**Strengths:**

- This paper addresses social navigation with collision avoidance and includes a complete policy—training pipeline.

- Under the success criterion, the proposed method achieves an increase in some metrics, such as episode success rate, finding success.

- By building upon SDA-style dynamic perturbations, the learned policy exhibits improved
generalisation in scaled and dynamic social scenarios.

- The work demonstrates that the plug-in communication module delivers increments when
integrated into the existing framework, validating its utility for social-navigation tasks and
obstacles avoiding.

**Weaknesses:**

- The absolute value of the Episode-Success(ES) metric is still low (~24%), leaving limited headroom
to showcase the true benefit of the communication augmentation.

- A comparative discussion against other relevant social navigation or generic multi-agent baselines
is missing, hindering an assessment of relative advancement.

- Evidence could be provided on transference from Habitat-Sim to a real-world platform; the gap
between simulation and physical deployment remains unaddressed.

**Questions:**

- Bounding-Box Representation. The use of axis-aligned bounding boxes for humans may overapproximate occupancy and discard posture detail. What is the sensitivity to sensor noise or
missed detections, and are there plans to adopt part-based or mesh-level representations?

- Distributed Computing Cost. The privileged-information description implies a single-agent/singlehuman setup. In the multi-agent field deployment, does each robot require a full CommNet stack,
and what is the associated on-device compute overhead?

- Training-Step Anomaly. In Appendix B, increasing steps from 200 M to 270 M raises the average
steps-to-first-success for the Comm-enabled policy (608 → 618), whereas the no-Comm
baseline improves (618 → 573). Please comment on this counter-intuitive trend: could it indicate
over-fitting to communication cues or reward mis-alignment?

- Platform Selection Rationale. Among Isaac Sim, AI2-THOR, Gibson, etc., what criteria led to the
choice of Habitat-Sim? And for the Habitat 3.0c, is there a selective-communication scheme in
Habitat 3.0c to reduce redundant queries?

- State-Space Organisation and Alignment. Beyond encoding the target estimate, what mechanism
prevents redundancy or misalignment among the S-vector components. Does the pipeline perform
any per-step semantic calibration with the interlocutor to mitigate drift (cf. Table 5 ablation)?

---

### Official Review · Reviewer_LXSs · 2025-10-30

**Soundness:** 3
**Presentation:** 3
**Contribution:** 2
**Rating:** 4
**Confidence:** 3

**Summary:**

This paper introduces a framework that enables robots to navigate through social dialogue with humans presented in the environment by leveraging LLMs. The proposed system consists of a language interaction module and a social behavior planning module, which together allow communication-enabled navigation. The framework is evaluated through simulation experiments.

**Strengths:**

1. The idea of allowing robots to interact with humans for directions is intuitive and practically meaningful. The authors successfully formulate this natural interaction into a scientific problem and propose a new framework to address it.
2. The ablation study is well designed and shows that the proposed communication module and its communication inputs contribute to improving social navigation performance.

**Weaknesses:**

1. While the idea is conceptually strong and points toward an important direction for integrating HRI and LLM-based reasoning, the overall system maturity remains at an early stage. The experiments are purely simulated, and the interactions are simplified.
2. The technical description of each component seems insufficient. For instance, the paper does not clearly specify: 1) how the language queries are generated, 2) how human responses are formatted and transformed from natural language to actionable commands, or 3) how the behavior planner defines its reward function and integrates with the communication policy. Also, although a high-level framework diagram is provided, the flow of information (e.g., LLM -> Policy -> Action) is not explicitly visualized or discussed in sufficient depth.
3. This reviewer believes that the baselines are limited to non-communicative social navigation models (DDPPO, SDA), which only demonstrate that "adding communication" improves over "no communication". Also, although the paper introduces an LLM-based framework, it did not compare it with other LLM-based interactive agents introduced in literature.
4. The paper assumes human responses to be relatively simple and reliable, but in reality, human behavior can be highly dynamic, ambiguous, or even misleading. Also, it is unclear how human feedback is modeled, whether it is rule-based, scripted, or simulated.
5. Without real-world experiments, the proposed framework remains only partially validated. Real human-robot interactions are inherently uncertain, and additional experiments involving real participants would be essential to confirm robustness and practicality.

**Questions:**

1. How does the proposed framework address cases where the human provides inaccurate or noisy information?
2. Similarly, how would the framework handle non-cooperative human responses? For instance, if two people provide contradictory guidance, how does the system resolve it?
3. Since the baselines are non-communicative models, do the authors think the results mainly show that communication is useful in general, rather than demonstrating the specific advantage of your proposed framework?
4. In real-world settings, would the robot mechanically asking every person for information be socially acceptable? Could this lead to inefficient or even impolite interactions? Have the authors considered a selective or context-aware communication policy?
5. Was the QWEN3-8B model fine-tuned for the CommNav domain or used zero-shot via prompt engineering? If zero-shot, how consistent are the generated responses?

---

### Official Review · Reviewer_pe2Q · 2025-10-31

**Soundness:** 3
**Presentation:** 2
**Contribution:** 1
**Rating:** 2
**Confidence:** 4

**Summary:**

This paper presents a pipeline that uses an LLM to enable a robot to request information about an object from a human and then leverage that guidance to localize and navigate to the object using an RL policy. The approach is evaluated in an extended Habitat 3.0 simulation environment.

**Strengths:**

+ Leveraging LLMs for robot navigation and human-robot interaction represents a frontier research direction in robotics, and current community progress demonstrates promising potential.

+ This work develops a communication module to the Habitat 3.0 simulation environment, which could benefit the community if released as open source.

**Weaknesses:**

-  The novelty appears limited. The primary contributions seem to focus on prompt design and integrating a communication module into a standard DDPPO framework. Prompt design has already been extensively studied in many areas, especially in language-based human–robot interaction, including its application to robot navigation.

-  The human–robot interaction community has extensive prior work on robots asking and answering questions, as well as object reference (e.g., for object localization and navigation). A large body of recent research leverages language-based interactions driven by LLMs. However, this paper largely ignores relevant state-of-the-art work from the robotics community.

-  It is unclear why the problem is framed as a multi-agent navigation problem, given that the paper focuses on enabling navigation for a single robot rather than coordination among multiple agents.

- Real human responses can be incomplete or ambiguous. The paper does not explain how the system handles such cases. Similarly, the paper assumes that “multiple humans … are willing to help,” but if multiple responses are provided, potential conflicts between them are not addressed.

-  Social navigation has well-known challenges related to the sim-to-real gap. The paper does not discuss how the proposed approach can be deployed on physical robots in real-world environments.

-  Both language-based human–robot communication and social navigation require real-time performance. What is the runtime and latency of the approach?

**Questions:**

See Weaknesses

---

### Official Review · Reviewer_yRN4 · 2025-11-01

**Soundness:** 1
**Presentation:** 1
**Contribution:** 1
**Rating:** 2
**Confidence:** 4

**Summary:**

This paper presents an approach for a robot to proactively ask questions to bystanders in an environment when tasked to search for a specific human in an environment.

**Strengths:**

Interesting formulation for incorporating question-answering to find someone (or something). However, see below re. the concern about the claim of social navigation.

**Weaknesses:**

- The problem being solved is not clear. The paper seems to conflate social navigation (navigating among humans in a socially accpetable manner) with person-finding (searching for a specific human among a crowd).
- The proposed solution would allow a robot to search for a specific  person, but it would not allow it to improve its ability at *social navigaion*.
- The types of input sought from the human seem unrealistic to expect of a regular person. Humans are not god at estimating metric range values to arbitrary objects, or accurate cardinal bearing registration.

**Questions:**

How does this problem cover social navigation?

---

### Note · Authors · 2025-11-18

I have read and agree with the venue's withdrawal policy on behalf of myself and my co-authors.